# Development of a High-Resolution Acoustic Sensor Based on ZnO Film Deposited by the RF Magnetron Sputtering Method

**DOI:** 10.3390/ma14226870

**Published:** 2021-11-14

**Authors:** Dong-Chan Kang, Jeong-Nyeon Kim, Ik-Keun Park

**Affiliations:** 1Graduate School of Nano IT Design Fusion, Seoul National University of Science and Technology, 232, Gongneung-ro, Nowon-gu, Seoul 01811, Korea; dongchan@seoultech.ac.kr; 2Ginzton Laboratory, Stanford University, Stanford, CA 94305, USA; jnk1@stanford.edu; 3Department of Mechanical and Automotive Engineering, Seoul National University of Science and Technology, 232, Gongneung-ro, Nowon-gu, Seoul 01811, Korea

**Keywords:** high-resolution scanning acoustic microscope (HR-SAM), zinc oxide film, piezoelectric ceramics, acoustic sensor

## Abstract

In the study, an acoustic sensor for a high-resolution acoustic microscope was fabricated using zinc oxide (ZnO) piezoelectric ceramics. The c-cut sapphire was processed into a lens shape to deposit a ZnO film using radio frequency (RF) magnetron sputtering, and an upper and a lower electrode were deposited using E-beam evaporation. The electrode was a Au thin film, and a Ti thin film was used as an adhesion layer. The surface microstructure of the ZnO film was observed using a scanning electron microscope (SEM), the thickness of the film was measured using a focused ion beam (FIB) for piezoelectric ceramics deposited on the sapphire wafer, and the thickness of ZnO was measured to be 4.87 μm. As a result of analyzing the crystal growth plane using X-ray diffraction (XRD) analysis, it was confirmed that the piezoelectric characteristics were grown to the (0002) plane. The sensor fabricated in this study had a center frequency of 352 MHz. The bandwidth indicates the range of upper (375 MHz) and lower (328 MHz) frequencies at the −6 dB level of the center frequency. As a result of image analysis using the resolution chart, the resolution was about 1 μm.

## 1. Introduction

Typical methods used to characterize micro-nanomaterials as well as detect defects and discontinuities on the surface of and inside materials include X-ray diffraction (XRD) and scanning electron microscopy (SEM). These non-destructive methods often require complex sample preparation, including specimen collection, surface treatment, and a controlled vacuum environment. Non-destructive methods are used to obtain surface structures of thin films and subsurface micro-faults, and the material properties are limited.

Acoustic microscopy uses the elastic properties of the material to construct images of and quantitively characterize them. The most important part of an acoustic microscope is an acoustic sensor—consisting of an acoustic lens and piezoelectric element—which transforms electrical signals into acoustic signals and vice versa. The acoustic sensor is designed and fabricated with a piezoelectric ceramic on one end to generate ultrasound waves and a lens on the other end to focus the waves on a specimen [1].

Currently, the most widely used piezoelectric ceramics are lead zirconate titanate PbZr_0.52_Ti_0.48_O_3_ (PZT)-based due to their excellent piezoelectric properties, and they are widely used in piezoelectric igniters and piezoelectric ultrasonic sensors. However, PZT-based ceramics contain more than about 60% lead(II) oxide (PbO), which is an inorganic compound detrimental to the human body in addition to being hazardous and toxic to the environment. For this reason, many countries around the world have recently restricted the use of Pb. Alternatively, research on zinc oxide (ZnO), aluminium nitride (AlN), cadmium sulfide (CdS), cadmium selenide (CdSe), and hafnium oxide (HfO_2_) has been actively conducted to find lead-free piezoelectric ceramics [2,3,4,5,6].

ZnO is classified as a group II-VI semiconductor, and is a material used in electronic, optoelectronic, and laser technology due to its wide energy band, high bonding energy, and high thermal as well as mechanical stability [7,8]. In addition, it is attracting attention in the ceramic industry due to its hardness, rigidity, and piezoelectric constant, and has low toxicity, biocompatibility, biodegradability, biomedicine, and a pro-ecological system [9,10]. ZnO piezoelectric ceramics—among various piezoelectric thin-film materials—have a wurtzite crystal structure with permeability and a high refractive index in the visible-light spectrum, a large piezoelectric constant, and a high electromechanical and nonlinear electro-optical coefficient. ZnO piezoelectric ceramics tend to resonate at nearly 1/4 wavelengths [11] and are used in various scales of applications: SAW filters, microcantilevers, transparent electrode materials, biosensors, etc. [12,13,14,15].

ZnO thin films have a high coupling coefficient, and its preferential growth orientation is in the c-axis into a wurtzite structure with hexagonal units [16,17,18,19,20]. Methods to fabricate ZnO thin films are metal organic chemical vapor deposition (MOCVD) [21,22], molecular beam epitaxy (MBE) [23], electrodeposition [24], radio frequency (RF) magnetron sputtering [17,18,19,20,21,22,23,24], sol–gel [25], and microwave synthesis [26]. Among these fabrication methods, RF magnetron sputtering uses simple equipment to grow ZnO thin films using a reactive gas mixture of argon and oxygen on ZnO targets. The deposition of ZnO thin films using RF magnetron sputtering has the advantages of obtaining transparent, dense, high-quality thin films with a high deposition rate, wide deposition area, and superior directionality (preferential orientation on the c-axis) [23,24]. The wurtzite crystal structure has a high electromechanical coupling coefficient and high piezoelectric properties. The ZnO thin film and the c-cut sapphire in this transducer have an epitaxial growth relationship with nearly no lattice mismatch, enabling the fabrication of a high-quality transducer [16,17,18,19].

The higher the frequency of the acoustic sensor, the higher the resolution, but the shorter wavelength, higher attenuation, and shorter focal length limit subsurface analysis. For this reason, the frequency cannot be increased indefinitely, and must be designed and manufactured according to the application.

ZnO can be deposited in a variety of ways. However, not only the (0002) plane with the best piezoelectric properties but also the (103) plane and the (004) plane grow. The growth direction may vary depending on various conditions, such as film thickness, temperature, gas flow, pressure, working distance, and substrate [27,28]. 

In this study, we found the optimal growth conditions for ZnO films using RF magnetron sputtering. An acoustic sensor for high-resolution acoustic microscopy with sub µm resolution was fabricated using ZnO piezoelectric ceramics, which can be used for surface microstructure analysis and subsurface inspection of micro/nano scale thin-film structures.

## 2. Acoustic Sensor Theory

Ultrasound propagates through an optically opaque material and is sensitive to elastic properties, densities, and grain sizes of propagation mediums. A scanning acoustic microscope (SAM) uses ultrasound waves and can detect properties as well as construct images of discontinuities on the surface and subsurface of a material [29,30,31,32].

Figure 1 shows the shape of an acoustic sensor, which consists of a piezoelectric ceramic that transmits and receives sound, and a crystal rod, also known as buffer rod, that acts as a waveguide. When a signal is sent to the piezoelectric ceramic, ultrasound waves corresponding to the input signal frequency are generated and transmitted through the crystal rod. The crystal rod is made of sapphire, since it has good acoustic transmission properties and can transmit sound waves to a sample with high acoustic velocity. Additionally, it reduces spherical aberration when the waves are focused by a lens. The lens is processed on the backside of the crystal rod in the form of a concave sphere to focus the incident waves locally on a specimen. To form an acoustic path between the acoustic sensor and the sample, a couplant-typically water is used. The propagation of ultrasound is more effective in water due to its higher characteristic acoustic impedance (1.48 MRayl) than that of air (415 Rayl). In order to minimize the propagation loss, which increases with the square of the frequency, a shorter focal length should be achieved.

## 3. Materials and Methods

The E-beam evaporation system (NEE-4000, NANO-MASTER, Austin, TX, USA) is used to deposit Ti and Au thin films, which were used as the upper and lower electrodes in this study. Au, as an electrode, does not react with other chemicals, and has a high electric conductivity. A Ti thin film was deposited as an adhesive layer to improve the bonding characteristics of Au. Using the E-beam evaporator equipment, the upper and lower electrodes were fabricated under the following deposition conditions: 2.0 × 10^−6^ torr for the base pressure, 25 °C for the temperature, and about 50 cm for the distance between the source and the substrate. The E-beam evaporator deposition speed was 0.5 Å/s for Au, and 0.1 Å/s for Ti.

The rf magnetron sputtering was processed by turning the plasma off three times to lower the temperature inside the chamber to avoid damaging the equipment while depositing 4–5-micrometer-thick ZnO film. It also helped prevent recrystallization and changes of the crystal growth plane of the ZnO thin film during the deposition. A high-purity (>99.99%) ZnO target with a 2-in diameter and 4 mm thickness was used for the sputtering. The conditions of the depositions were as follows: initial degree of vacuum was 7 × 10^7^ torr, the pressure of the process chamber was 10 mTorr, the deposition at discharge power of 400 W, and the Ar/O_2_ gas mixture ratio was 30:10 sccm. The deposition speed of RF magnetron sputtering was 23 Å/s.

The c-cut sapphire rod of the high-resolution acoustic sensor was designed as seen in Figure 2. C-cut sapphire is a widely accepted material as a buffer rod for its high sound speed (11,175 m/s longitudinal and 6950.0 m/s shear wave velocities) [33]. The ultrasound-generating piezoelectric ceramics are deposited on the upper part of the buffer rod, which is why the surface of the upper part must be processed with precision. Additionally, the stress generated during the cutting and polishing processes must be minimized to lower the probability of exfoliation of the depositing thin films as deposition of piezoelectric ceramics proceeds in the vacuum equipment.

The diameter of the c-cut sapphire buffer rod is designed to be larger than that of the transducer to avoid the side wall reflections. The length of the buffer rod must be longer than the near-field (Fresnel zone) distance, which is calculated using Equation (1):(1)$$N=D^{2}/4\lambda_{\mathrm{rod}}$$
where N refers to the near-field (Fresnel zone) distance, D is the diameter of the piezoelectric element, and λ_rod_ is the wavelength of the material used for the buffer rod. The ultrasound travelling along the buffer rod should be focused at the end of the lens on which the lens aperture with a curvature is placed to focus the ultrasound waves on a specimen. The width of an aperture, the aperture angle, and the radius of the curvature are related as: (2)$$r=R\;\sin\left(\frac{\theta_{\alpha }}{2}\right)$$
where r is half of an aperture width, R is the curvature radius of the aperture, and $\theta_{\alpha }$ is an aperture angle. A focal distance of the aperture can be expressed in terms of the ratio of longitudinal wave velocities of the adjacent medium, and the radius of the curvature as: (3)$$F_{0}=\frac{R}{\left(1-C_{2}/C_{1}\right)}$$
where $C_{1}$ and $C_{2}$ are the longitudinal velocities of the buffer rod and the coupling media, respectively. The designed lens is shown in Figure 2, and the dimensions are listed in Table 1.

In determining the thickness of the piezoelectric element for a desired frequency, the acoustic impedances of the buffer rod and the piezoelectric element must be considered [34]. If the acoustic impedance of the buffer rod is greater than that of the piezoelectric element, the thickness of the piezoelectric element is determined to be λpiezo/4, where λpiezo is the wavelength of the piezoelectric element. The piezoelectric element thickness should be λpiezo/2 for cases where the piezoelectric element has a larger acoustic impedance than that of a buffer rod. The acoustic impedances of sapphire and ZnO are 44.3 MRayl and 36.4 MRayl, respectively, and the acoustic velocity of the ZnO is about 6400 m/s. Table 2 shows the thickness of the deposited piezoelectric element along with its diameter and those of the upper and lower electrodes. Figure 3 displays the schematic of the ZnO film acoustic sensor of this study.

## 4. Result and Discussion

Figure 4 shows the X-ray diffraction (XRD, D8 Discover with GADDS) graph of the 4.87-micrometer-thick ZnO film deposited by RF magnetron sputtering. The measurement conditions for the XRD were 2°/min scan and in the 2θ = 20°–80° range. The peak occurred at about 34°, indicating that there was growth in the (0002) direction. It was confirmed that the inflow of excess oxygen resulted in the growth of a wurtzite crystal with the c-axis preferential orientation. To evaluate the grain size of ZnO at this location, the full width at half maximum (FWHM) was measured using the ω-scan of XRD in the crystal growth direction (0002). The FWHM obtained by the ω-scan of XRD was 0.262°, which was excellent in crystallinity, and the piezoelectric conductivity is known to increase as the value of the FWHM of piezoelectric ceramics decreases. Using the peak positions and FWHM values, the Scherrer equation (Equation (4)) can be applied to determine the crystallite or particle size:(4)$$\tau =\frac{K\lambda }{\beta \cos\theta }$$
where τ is the mean size of the crystalline, K is the constant, λ is the X-ray wavelength, β is the FWHM, and θ is the Bragg angle. Using the XRD results, the average crystallite size is about 33.15 nm.

Figure 5 shows the microstructure of the ZnO film seen through a field emission scanning electron microscope (FE-SEM, JEOL-JSM-6700F, Tokoy, Japan), where particles of 100~200 nm grew relatively densely and regularly. SEM and XRD determine different kinds of sizes. SEM allows physical particles to be seen and XRD allows the size of the crystalline domains to be calculated. It is known that XRD can be used to determine the average column height inside the crystallites (this value is often referred to as the ‘crystallite size’), and sometimes the column height distribution [35]. Studies have shown that the crystallite sizes obtained using XRD are about 100–200 nm smaller compared to SEM [36]. The crystallite size is calculated from XRD, whereas the particle size can be known by SEM.

Figure 6 shows the thickness of the cross-section of the deposited piezoelectric element using a focused ion beam (FIB, Nova 600 NanoLab, Pittsburgh, PA, USA). Figure 6 shows the cross-section of the (Figure 6a) deposited upper electrode, where the Au thin-film thickness is 246 nm and the Ti adhesion layer thickness is 92 nm, (Figure 6b) the ZnO piezoelectric thickness is 4.87 μm, and (Figure 6c) the lower electrode, were the Au thin-film thickness 122 nm and the Ti adhesion layer thickness 16 nm. 

Since the thickness of the thin film is thick, there is a possibility that the temperature of the sputtering chamber may rise, so there is a cooling time. Although the coefficient of thermal expansion of the ZnO film is relatively low, at 4 × 10^−6^ K^−1^, it is being studied that the growth direction changes when it exceeds about 300 °C [37,38]. Since the temperature rise due to plasma is inevitable due to a long sputtering operation, a cooling time to lower the temperature inside the vacuum chamber is essential. However, In Figure 7, it is shown that c-axis growth is successful in regards to the constant growth direction of the ZnO film.

Figure 8 shows the fabrication process of the sensor by depositing the piezoelectric ceramics on the sapphire lens and proceeding with a casing process. The sensor was fabricated by connecting it to the connector through Au wire bonding. Heat treatment was performed at about 200 °C using epoxy to fix the Au wire bonding, sensor case, and connector.

Figure 9 shows the pulse-echo signal response of the fabricated acoustic sensor. It is A-scan data obtained using a Si wafer as a reflector, and the distance between the acoustic sensor and the reflector was about 300 µm. Figure 10 shows the (fast Fourier transform) FFT of the obtained pulse-echo signal, which shows that the center frequency is about 352 MHz. The bandwidth indicates the range of the upper (375 MHz) and lower (328 MHz) frequencies at the −6 dB level of the center frequency. The −6 dB bandwidth of the manufactured acoustic sensor was calculated as {(375 − 328)/352} × 100% = 13.35%.

The resolving power refers to the ability to distinguish two adjacent lines from each other, and is mainly used to check the performance of optical devices. For our resolution chart, PE-oxide (silicon oxide layers deposited by PECVD using SiH_4_ and N_2_O) was deposited on a silicon wafer using CVD (chemical vapor deposition) equipment, and the pattern was fabricated using a mask with constant lines with a thickness from 30 µm to 1 µm. Figure 11 shows images of the resolution chart: (a), (c), and (e) are optical microscope (OM, OLYMPUS BX51-P, Ontario, NY, USA) images, and (b), (d), and (f) are images obtained by an ultrasonic microscope using the fabricated sensor. Figure 11a,b is 15-micrometer- and 1-micrometer-thick lines, Figure 11c,d is 10-micrometer- and 1-micrometer-thick lines, and Figure 11e,f is 5-micrometer- and 1-micrometer-thick lines, respectively. When observed through the ultrasonic microscope using the sensor we fabricated, a 1-micrometer-thick line appeared clearly.

E-beam evaporation is a typical vacuum thin film deposition system of the physical vapor deposi-tion (PVD) method, which is a thin film fabrication method using physical methods, and its schematic diagram appears in Appendix A [39].

RF magnetron sputtering method was used for the ZnO thin film deposition, the most fundamen-tal part of the acoustic sensor. A schematic of the RF magnetron sputtering equipment is shown in Appendix A [40]. In Appendix A, MFC (Mass Flow Controller) is a controller to precisely control the amount of injected gas. The unit is sccm (standard cubic centimeter per minute).

## 5. Conclusions

In this study, ZnO piezoelectric ceramics with a high-resolution acoustic frequency were fabricated using RF magnetron sputtering and E-beam evaporation equipment. A ZnO film with a c-axis preferential orientation was deposited using RF magnetron sputtering, and its thickness was measured to be about 4.87 µm. For optimal growth of the (0002) plane of the ZnO film, an appropriate cooling time was required to maintain the sputtering temperature below 300 °C.

The upper and lower electrodes of Ti and Au thin films were fabricated using E-beam evaporation, and Ti was used as an adhesion layer for the Au thin films with poor bonding properties. The thickness of the Ti and Au thin films at the lower electrode was measured to be 16 nm and 122 nm, respectively. In the upper electrode, they were observed as Ti 92 nm and Au 246 nm, which were close to the sensor design. It is judged that the difference in the thickness of the piezoelectric thin film is different from the design value because it is difficult to keep the voltage and current values constant.

The sensor fabricated in this study had a center frequency of 352 MHz, and it showed that it had a resolving power of about 1 μm upon image analysis using the resolution chart. The piezoelectric ceramic properties of the ZnO film, which become thinner with increasing frequency, confirm the feasibility of fabricating high-resolution acoustic sensors with higher frequencies.

## Figures and Tables

**Figure 1 materials-14-06870-f001:**
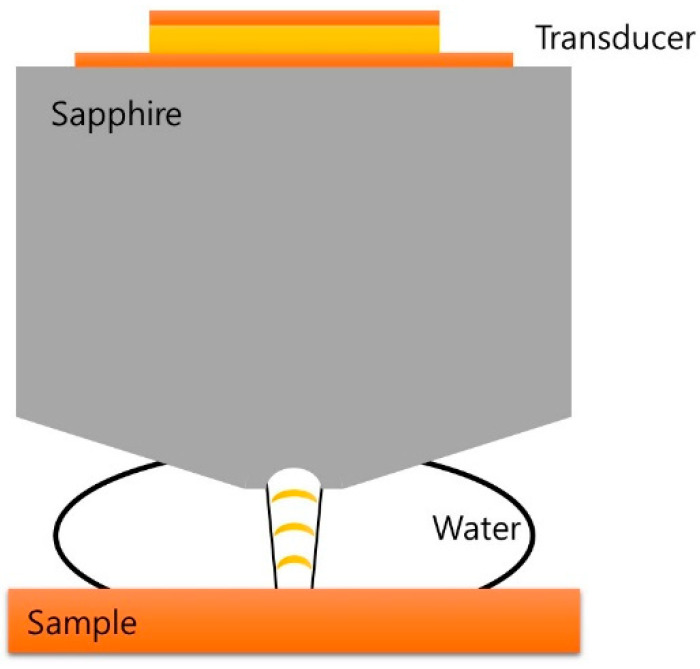
Schematic diagram of ultrasonic reflective focus-type acoustic sensor.

**Figure 2 materials-14-06870-f002:**
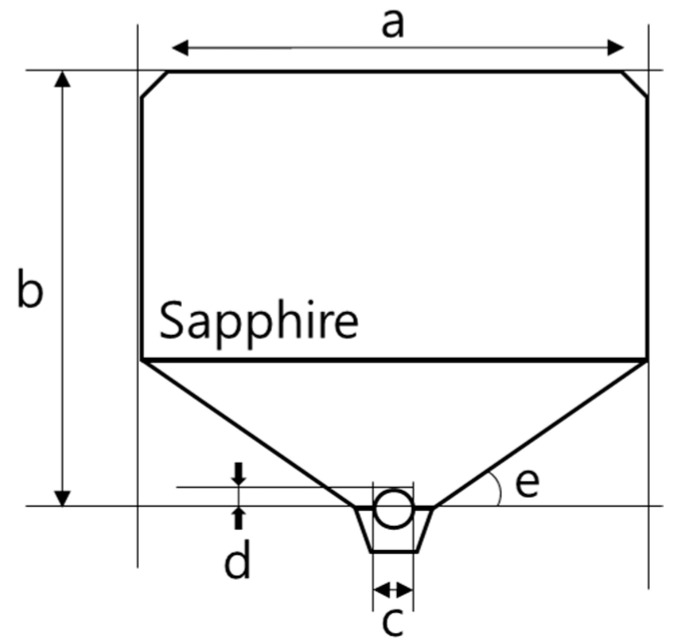
Schematic design cross-section of the sapphire buffer rod.

**Figure 3 materials-14-06870-f003:**
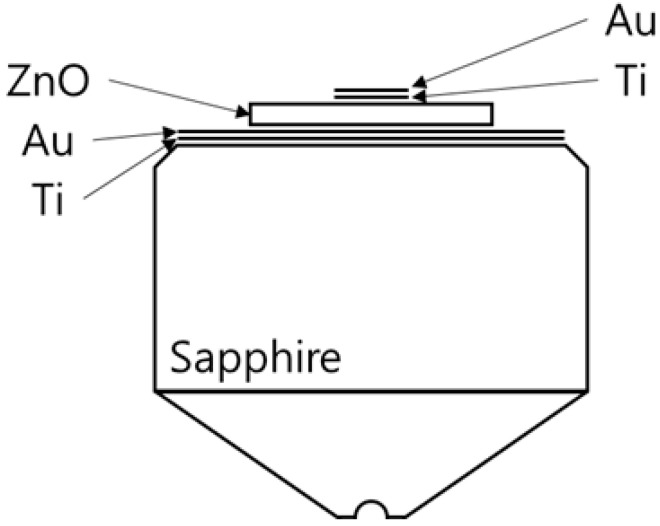
Schematic drawing of the ZnO film transducer structure.

**Figure 4 materials-14-06870-f004:**
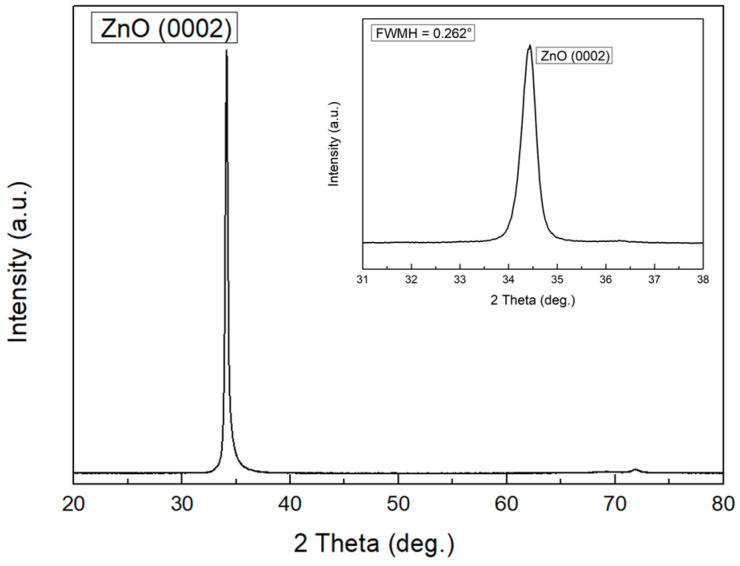
X-ray diffraction graph of ZnO film grown with (0002) plane deposited by RF magnetron sputtering.

**Figure 5 materials-14-06870-f005:**
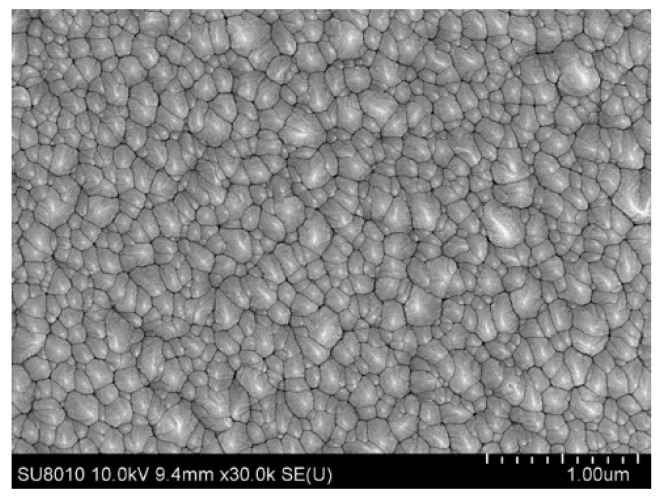
Scanning electron microscope micrographs of ZnO film.

**Figure 6 materials-14-06870-f006:**
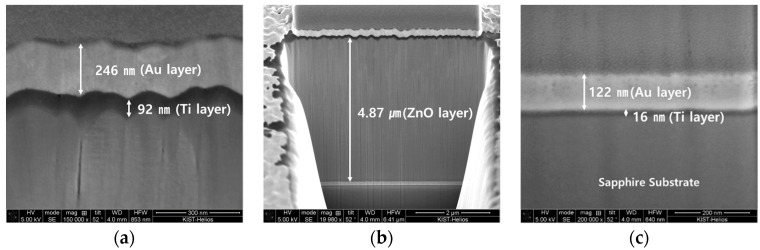
Piezoelectric FIB thickness images. (**a**) Image of the deposited top electrode at 246 nm for the Au layer and 92 nm for Ti layer. (**b**) Image of the deposited ZnO transducer at 4.87 µm. (**c**) Image of the deposited bottom electrode at 122 nm for theAu layer and 16 nm for the Ti layer.

**Figure 7 materials-14-06870-f007:**
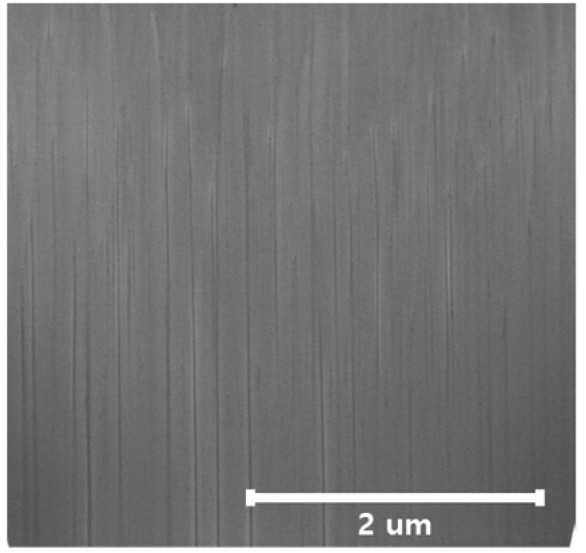
FIB cross-sectional image of a ZnO film grown along the c-axis.

**Figure 8 materials-14-06870-f008:**
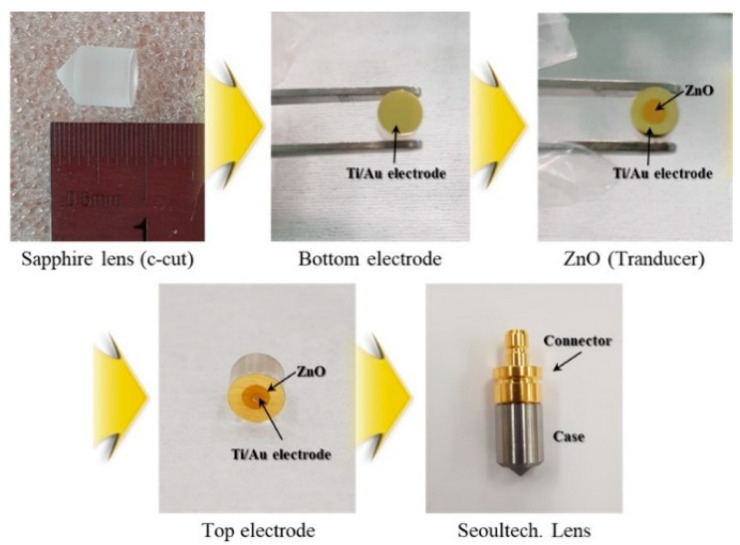
Image of fabrication process of the ZnO piezoelectric transducer.

**Figure 9 materials-14-06870-f009:**
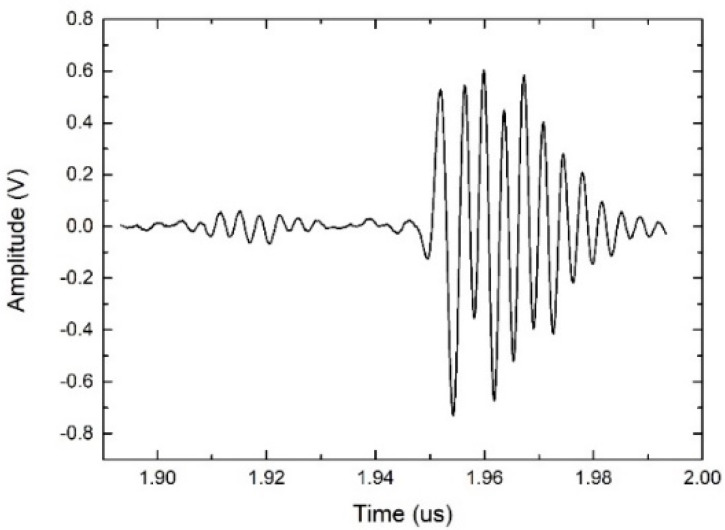
Measured pulse-echo response spectrum of the fabricated transducer.

**Figure 10 materials-14-06870-f010:**
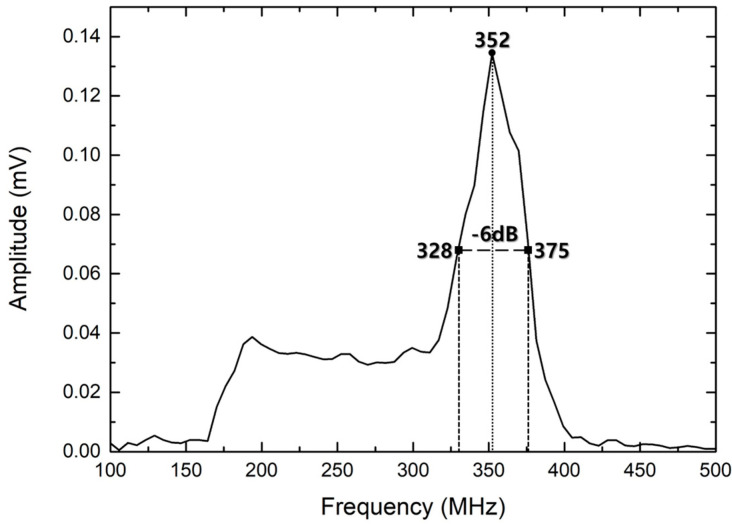
Measured fast Fourier transform (FFT) and bandwidth spectrum of the transducer.

**Figure 11 materials-14-06870-f011:**
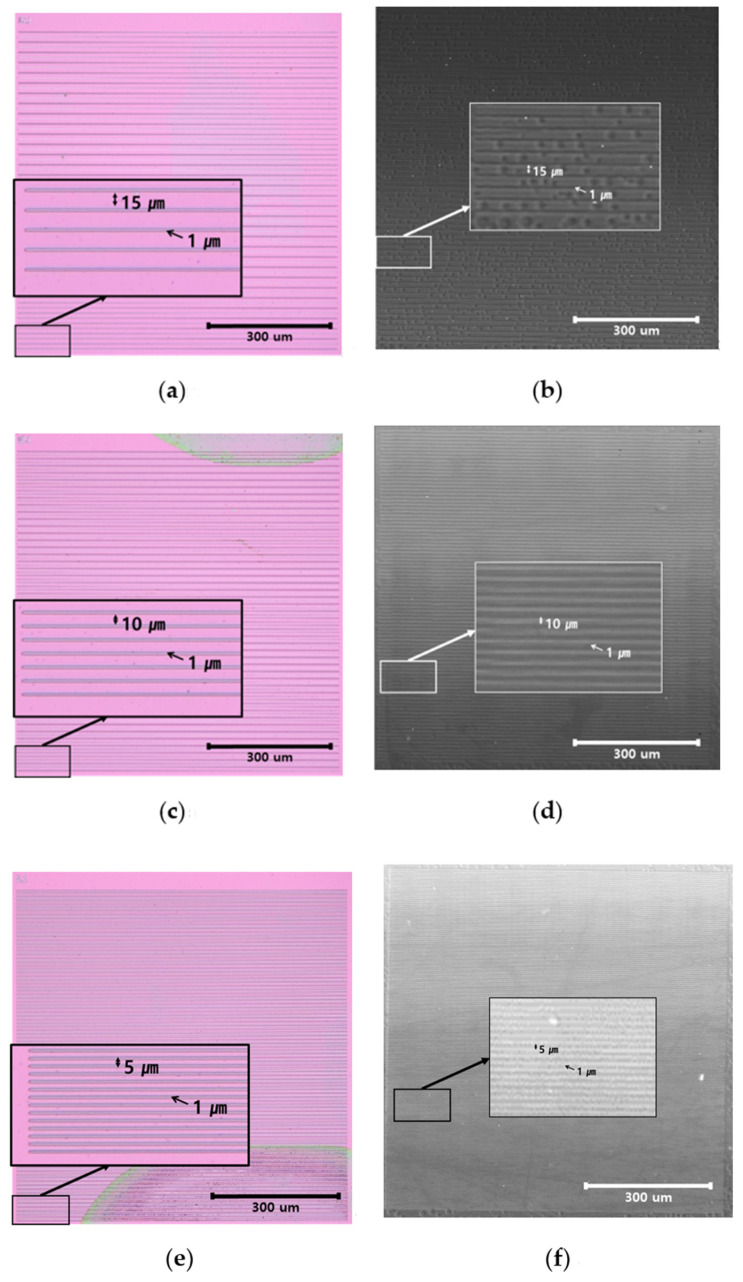
Calibration block for the verification of lateral resolution measurements. Optical microscopy (left column) images and fabricated acoustic sensor (right column) images of the calibration block are composed of equally spaced lines. (**a**,**b**) are 15-micrometer- and 1-micrometer-thick lines, (**c**,**d**) are 10-micrometer- and 1-micrometer-thick lines, and (**e**,**f**) are 5-micrometer- and 1-micrometer-thick lines.

**Table 1 materials-14-06870-t001:** Designed Sapphire Buffer Rod Diameters (mm).

	a	b	c	d	e
Diameter	6.1 ± 0.005	6.45 ± 0.005	0.866 ± 0.005	0.25 ± 0.005	40°

**Table 2 materials-14-06870-t002:** Material Thickness and Diameter Design of Piezoelectric/Electrode Films.

	Bottom Electrode		Transducer	Top Electrode	
Material	Ti	Au	ZnO	Ti	Au
Thickness	10 nm	150 nm	5.1 µm	50 nm	250 nm
Diameter	6 mm	6 mm	3 mm	1 mm	1 mm

## Data Availability

Not applicable.

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
