# Peer review of "Development of a High-Resolution Acoustic Sensor Based on ZnO Film Deposited by the RF Magnetron Sputtering Method"

_materials, 2021, doi:10.3390/ma14226870_

Round 1

Reviewer 1 Report

The authors report on the building and the characterizations of an acoustic sensor for a high-resolution acoustic microscope. For this, they used piezoelectric ZnO as well as sapphire materials as principal constituents of their microscope. This microscope could be of utmost of importance in material science and engineering but its construction remains very basic.
While the study is clear, the whole paper suffered a lot of aim, challenge and even context. For instance, it is not expressed in the introduction the survey of such technique in literature ... The  importance of these results are also not discussed. For this reason, I do not recommend this publication in this form in a journal focused on materials. A more technology journal should be targeted.

1. Please, provide also the specular scan and not only the rocking curve scan.
2. Line 14, C capital should be replaced by c.
3. Line 18, Sapphire in capital should be replaced by sapphire.
4. The authors should describe better the acoustic measurement in the abstract.
5. In the keywords, the authors said HR-SAM while it's not reported once in the core of the study. Acoustical should also be replaced by acoustic.
6. Line 29, XRD and SEM are nondestructive?! The introduction should be rephrased.
7. Other piezo/ferroelectric material are used, as for instance HfO2 and its cousins. It should be included in the study.
8. As it can be found in literature, RF sputtering is one of the worst techniques for the growth of high piezoelectric coefficient ZnO. It induces different sizes of grain and so on. I would not provide any literature since I also published on this topic.
9. The authors should express the depth sensitivity of this technique.
10. The authors should use the same terms present in the equations and in the table 1.
11. Line 145, [x] stands for?
12. In the table 2, the authors express a ZnO thickness of 5.1 um while it's written 4.87 all over the manuscript.
13. PE-Oxide stands for?
14. In Figure 4 and 5, sapphire had a misspelling.
15. If the authors want to keep figure 7 and 8, they should describe more their analysis. The figure 7 should be placed in the supplementary material.
16. The figure 11 should be shown in frequency and not in time.
17. Can the authors describes more the defects observed in Figure 13 (a) and (b) ?
18. The figure 2 and 3 should be placed in supplementary. Everyone knows the principle of e-beam as well as sputtering.
19. All the legends of figures should be described more.
20. Speaking about piezoelectric material without any piezoelectric measurements is always tricky ...

Reviewer 2 Report

Dear Authors, in your interesting manuscript, the following points should be added/changed to further improve:

  1. Abstract: Please provide definitions for the abbreviation "RF".
  2. Introduction: I would suggest to the authors that they add one sentence of commentary stating that zinc oxide is a multifunctional material. Many authors forget to provide this relevant information in their works.
  3. Introduction: Comment on the sentence ” Methods to fabricate ZnO thin films are metal organic chemical vapor deposition (MOCVD) [16-17], molecular beam epitaxy (MBE) [18], electrodeposition [19], radio frequency (RF) magnetron sputtering [12-20], and Sol-gel [21]. (56-59)” I also suggest mentioning microwave synthesis, which is described in detail in the article (doi:10.3390/nano10061086).
  4. Introduction: Comment on the sentence “In this study, we fabricated an acoustic sensor for high-resolution acoustic microscopy using ZnO piezoelectric ceramics, which can be used for surface microstructure analysis and subsurface inspection in thin film structures. (68-70)” Please define the term 'high-resolution'.
  5. Materials and Methods: What purity substrate was used to produce thick ZnO film?
  6. Materials and Methods: Comment on the sentence “The conditions of the depositions were as follows: initial degree of vacuum was 7 × 10−7 torr, the pressure of the process chamber was 10 mTorr, the RF voltage was 400 W, and the Ar/O2 gas mixture ratio was 30:10. (111-113)” A comment on the "400W" value. There is a mistake in the sentence, whether the value 400 refers to voltage or power?
  7. Materials and Methods: Please provide definitions of the abbreviation MFC (Figure 3).
  8. Materials and Methods: The authors did not provide information on the SEM and XRD analyser used.
  9. Materials and Methods: The authors did not provide information on the optical microscope used.
  10. Result and Discussion: Why the authors did not determine the average crystallite size using XRD results?
  11. Result and Discussion: Please describe the films/materials in the images (Figure 8). Currently it's hard to find out what is what.
  12. Result and Discussion: Please add scale bar (Figure 9).
  13. Conclusions: Comment on the sentence “The thickness of the Ti and Au thin films at the lower electrode was measured to be 16 nm and 122 nm, respectively. In the upper electrode, they were observed as Ti 92 nm and Au 246, which were close to the sensor design. (217-219)” Please show me where in the manuscript the authors determined the thicknesses of Ti and Au thin films?
  14. Conclusions: Comment on the sentence “The thickness of the Ti and Au thin films at the lower electrode was measured to be 16 nm and 122 nm, respectively. In the upper electrode, they were observed as Ti 92 nm and Au 246, which were close to the sensor design. (217-219)” Please explain to me why the determined values do not agree with the data in Table 2?

Round 2

Reviewer 1 Report

The authors really improved the draft paper but still need some important revisions to fully convinced me of a consistent and valuable story. I try to addressed my points below plus additional comments.

  1. I would suggest to show the full 2Theta range. I was probably not clear enough, my mistake. I would suggest to point out that the FWHM in the figure is obtained in omega scan. I would suggest to describe more the capture legend saying for instance at which step of the process this XRD measurement was done.
  2. They are still some C-cut in capital though the paper, as for instance in line 140-141. Please, correct them.
  3. Ok.
  4. I now agree with the authors.
  5. Ok.
  6. I would simply avoid to begin the introduction like this because I don’t see why SEM or XRD are so complicated to do. And also because XRD and SEM measurements were done in this actual study.
  7. Ok.
  8. I fully agree that RF sputtering is an easy technique and well used method to deposit ZnO growth. I would just avoid to say that RF growth of ZnO gives high piezoelectricity, which is now the case.
  9. I asked for lateral resolution, so probably what the authors denote as axial.
  10. Ok.
  11. Ok.
  12. Ok.
  13. Ok.
  14. Ok.
  15. "Since the thickness of the thin film is thick, there is a possibility that the temperature of the sputtering chamber may rise, so there is a cooling time. In this case, the growth direction of the thin film may be changed.". Interesting, can the authors supply some references and explain more this effect?
  16. Ok.
  17. Ok.
  18. Ok.
  19. They still need way more descriptions. For instance, in Fig.1, the authors use only 5 words... They should describe these figures legends in such way than the story could be self-consistent with only the figures ...
  20. Well, it could be beyond the scope of this study. I would not recommend to perform any specific measurement for the proof of concept of this research.

  1. The grain size measured by SEM is way larger than the one obtained in XRD. Can the authors explain this obvious discrepancy?
  2. In Figure 6, the thicknesses are hardly readable. Please, change the figure accordingly.
  3. Since the goal of Figure 11 is to check the 1 um pattern, I would suggest to change the scale of the figure and do zooms in appropriate regions.
  4. I would suggest to not say sensor operating at 400 MHz since the frequency center is 352 MHz. Please, use this value instead. It creates too much confusion.
  5. To my opinion, the scope of this study, the challenges versus the existing systems and the novelty/improvements done should be described more in the abstract/introduction/conclusion. It is really important for the readers to know these points. This particular point is the more important along these comments.
  6. Actually, 5um is not exactly what we called thin films, i.e. nm thickness range. I would simply used films all over the manuscript.
  7. Since the scope of this study is to design and make a high-resolution sensor for acoustic microscope, I would highlight this technology already in the title of the paper. Standing like this, the title would mislead readers that will understand that this particular study consists to create of an acoustic sensor made of ZnO, which is not particularly challenging, except if the authors explained me and the readers the actual challenges.

Reviewer 2 Report

Point 14: Conclusions: Comment on the sentence “The thickness of the Ti and Au thin films at the lower electrode was measured to be 16 nm and 122 nm, respectively. In the upper electrode, they were observed as Ti 92 nm and Au 246, which were close to the sensor design. (217-219)” Please explain to me why the determined values do not agree with the data in Table 2?

Response 14: The deposition rates shown on line 110 and line 119 of Materials and Methods are from several experiments. Added on Conclusions line 249-251.

Lines 249-251:

It is judged that the difference in the thickness of the piezoelectric thin film is different from the design value because it is difficult to keep the voltage and current values constant

Answer: I note that the authors wrote “Table 2 shows the thickness of the deposited piezoelectric element along with its diameter and those of the upper and lower electrodes. (179-182)” and ” Material thicknesses and diameters of piezoelectric/electrodes films.(186)” It does not follow that these are design values. Please correct the description of Table 2 so that it does not mislead the reader.   

Good Luck!
